# Improvement of Oxidative Stability of Fish Oil-in-Water Emulsions through Partitioning of Sesamol at the Interface

**DOI:** 10.3390/foods12061287

**Published:** 2023-03-17

**Authors:** Zhihui Gao, Zhongyan Ji, Leixi Wang, Qianchun Deng, Siew Young Quek, Liang Liu, Xuyan Dong

**Affiliations:** 1College of Food Science and Engineering, Qingdao Agricultural University, Qingdao 266109, Chinaliuliangwh@163.com (L.L.); 2Oil Crops Research Institute of the Chinese Academy of Agricultural Sciences, Wuhan 430062, China; chunn2@163.com; 3School of Chemical Sciences, The University of Auckland, Auckland 1142, New Zealand; sy.quek@auckland.ac.nz; 4Riddet Institute, Palmerston North 4474, New Zealand

**Keywords:** sesamol, fish oil, O/W emulsion, oxidation, interface

## Abstract

The susceptibility of polyunsaturated fatty acids to oxidation severely limits their application in functional emulsified foods. In this study, the effect of sesamol concentration on the physicochemical properties of WPI-stabilized fish oil emulsions was investigated, focusing on the relationship between sesamol–WPI interactions and interfacial behavior. The results relating to particle size, zeta-potential, microstructure, and appearance showed that 0.09% (*w*/*v*) sesamol promoted the formation of small oil droplets and inhibited oil droplet aggregation. Furthermore, the addition of sesamol significantly reduced the formation of hydrogen peroxide, generation of secondary reaction products during storage, and degree of protein oxidation in the emulsions. Molecular docking and isothermal titration calorimetry showed that the interaction between sesamol and β-LG was mainly mediated by hydrogen bonds and hydrophobic interactions. Our results show that sesamol binds to interfacial proteins mainly through hydrogen bonding, and increasing the interfacial sesamol content reduces the interfacial tension and improves the physical and oxidative stability of the emulsion.

## 1. Introduction

Polyunsaturated fatty acids (PUFAs), which are rich in docosahexaenoic acid (DHA) and eicosapentaenoic acid (EPA), exhibit beneficial effects on human health, such as anti-inflammatory, anti-cancer, anti-depression, and vascular protective activities, and promote neurological development [1]. PUFAs have great potential as functional food ingredients and thus, have become a research hotspot for domestic and international scholars [2]. However, PUFAs, including DHA and EPA, are highly susceptible to oxidation during processing and storage because of high level unsaturated bonds [3]. The reaction products of lipid oxidation not only adversely affect the flavor, nutrient content, and shelf life of foods but can also even damage the health of consumers [4]. Lipid oxidation severely limits the development and application of PUFAs in functional emulsified foods. Therefore, understanding and preventing lipid oxidation in complex food systems is essential for the construction of PUFAs emulsion systems with good oxidative stability. An emulsion delivery system is a classical lipid carrier that can significantly improve the bioavailability and stability of PUFAs.

Using antioxidants, especially natural antioxidants, is the most direct and effective way to inhibit lipid oxidation in the food industry [5]. Natural polyphenols, comprising chemical structures with antioxidant activity (e.g., catechols and hydroxyl groups), can inhibit lipid oxidation by scavenging free radicals or chelating transition metal ions. Furthermore, natural polyphenols are safe to consume and have no side effects, which has made them popular in research and production [6]. Natural phenolic antioxidants, such as tea polyphenols, quercetin, and tocopherols, improve oxidative stability and extend the shelf life of emulsions [7,8,9].

Sesamol is a plant-derived monophenolic compound extracted from sesame seeds. Its antioxidant activity is derived from the phenolic group on the benzodioxole ring [10]. Sesamol is used as an antioxidant in food because of its excellent ability to inhibit lipid oxidation. Sesamol not only inhibits single lipid oxidation, but also lipid oxidation in complex systems [11,12]. For example, sesamol significantly improved the oxidative stability of lipids in beeswax organogel systems [13]. Additionally, sesamol inhibits lipid oxidation in sunfower oil-in-water (O/W) emulsion systems [14]. The DPPH radical scavenging rate of the same concentration of sesamol (2.5 µmol/g) was 2.56 times higher than that of 2,6-di-tert-butyl-p-cresol in lard after thermal induction at 180 °C for 80 min [15].

The interfacial region, which separates the oil phase from the aqueous phase, plays a key role in inhibiting lipid oxidation in emulsions [16]. Tocopherols interacting with whey protein isolate (WPI) through hydrophobic and electrostatic interactions, were adsorbed at the interface, where they significantly inhibited lipid oxidation in O/W emulsions of linseed oil [9]. Numerous polyphenols also reside at the oil–water interface via non-covalent bonds with adsorbed proteins [16]. Wang et al. found that sesamol effectively inhibited particle aggregation and lipid oxidation in protein-stabilized flaxseed oil-in-water emulsions, and hypothesized that sesamol molecules could adsorb on the surface of oil droplets and interact with emulsifiers to influence interfacial properties, thereby enhancing the stability of emulsions [17]. However, the interaction between sesamol and emulsifiers was not explored and the partitioning of sesamol in all phases of the emulsion was not clarified. Therefore, we expected to understand the relationship of antioxidant-emulsifier interactions, interfacial partitioning, and emulsion stability by further investigating the interaction and interfacial partitioning of sesamol in WPI-stabilized fish oil emulsions.

In this study, the effect of sesamol on the oxidative stability of WPI-stabilized fish oil O/W emulsions was investigated, focusing on sesamol interfacial partitioning and interactions. Furthermore, a preliminary study of its potential mechanism of action was conducted. The sesamol content in each phase was analyzed separately at different storage periods to elucidate the interfacial distribution of sesamol during lipid oxidation in emulsions. This study provides guidance for the application of sesamol as a plant-based antioxidant in functional foods, thereby providing a theoretical basis for the development of functional emulsified foods enriched with PUFAs.

## 2. Materials and Methods

### 2.1. Materials

Sesamol (98%) was purchased from Shanghai Yi’en Chemical Technology Co., Ltd. (Shanghai, China). WPI (92%) was purchased from Shanghai Yuanye Biotechnology Co., Ltd. (Shanghai, China). Fish oil was purchased from DSM Nutritional Products Ltd. (Kaiseraugst, Switzerland). Except for methanol, which was chromatographic grade, all other chemicals were analytical grade chemicals and were purchased from Sinopharm Chemical Reagent Co. (Beijing, China). Double-distilled, deionized water was used for the preparation of all solutions.

### 2.2. Preparation of Stripped Fish Oil

Stripped fish oil was prepared in order to exclude the interference of the original phenolics in the fish oil. The fish oil was processed based on the method described by Cheng et al. with minor modifications. The chromatographic silica was repeatedly rinsed with double-distilled water at room temperature until it was free of impurities and then activated at 120 °C for 12 h. Equal amounts of fish oil were dissolved in 100 mL of n-hexane, loaded onto the column, and then eluted with 10 times n-hexane (at room temperature). The glassware for collecting fish oil was wrapped with aluminum foil and put in an ice bath to avoid lipid oxidation during collection, and the collected mixture was placed in storage at −80 °C. Before use, n-hexane was removed using a vacuum rotary evaporator (IKA, RV10D, Staufen, Germany) at 37 °C, and the remaining solvent was evaporated with nitrogen [18].

### 2.3. Preparation of Fish O/W Emulsion

WPI (1% *w/v*) was dissolved in phosphate buffer solution (PBS, 10 mM, pH 7.0) to form the aqueous phase. The oil phase was prepared by adding sesamol (sesamol content in the emulsion was 0, 0.01, 0.03, or 0.09%, w/v) to the fish oil. The 5% (*v/v*) oil phase and 95% (*v/v*) water phase were mixed and processed by a high-speed stirring device (IKA, T25, Staufen, Germany) at 15,000 rpm for 2 min, followed by placing the primary emulsion on an ice bath and subjecting it to 650 W ultrasonication (Nanjing Xianou Instrument Manufacturing Co., Ltd., XO-1000D, Nanjing, China, amplifying bar Φ 6) for 5 min. Sodium azide (0.02% *w/w*) was added to the emulsions to prevent microbial growth.

### 2.4. Accelerated Storage Experiments

The emulsions were oxidized under the Fenton system for 5 days, with samples collected every 24 h for analysis. The Fenton system was generated from a recovered solution of 10 µM FeCl_3_, 100 µM ascorbic acid, and 5 mM H_2_O_2_ [19].

### 2.5. Measurement of Particle Size and ζ Potential during Storage of Emulsions

The laser particle size analyzer (Microtrac MRB, S3500, Montgomeryville, PA, USA) measured the particle size of the emulsion with refractive indices set to 1.49 (oil) and 1.33 (deionized water) [20]. The results were recorded as volume-weighted average particle size (D_4,3_):(1)$$D_{4,3}=\sum n_{i}d_{i}^{4}/\sum n_{i}d_{i}^{3}$$

*n_i_*: number of droplets, *d_i_*: droplet diameter.

The zeta potential was measured using dynamic light scattering (Malvern, NANO ZS90, Malvern, UK). The emulsion samples were diluted 200 times with PBS (10 mM, pH 7) at 25 °C and loaded into DTS1070 measuring dishes. The Hückel model was selected for the measurement calculation [21].

### 2.6. Observation of the Microstructure and Visual Appearance during Emulsion Storage

The microstructure of the emulsion was photographed using a confocal laser scanning microscope (Zeiss, LSM900, Oberkochen, Germany) equipped with a 60× oil immersion objective [22]. Nile Red (1 mg) was dissolved in 10 mL of ethanol, and 20 µL was added to 1 mL of emulsion to stain the oil phase. An Ar laser was used to excite the Nile red dye fluorescence at 488 nm.

### 2.7. Analysis of Lipid Oxidation during Emulsion Accelerated Storage

The extent of lipid oxidation was monitored by periodically measuring the amount of hydrogen peroxide and thiobarbituric acid reactive substances (TBARS) formed in the emulsion [23]. The method developed by Shantha and Decker was optimized to determine lipid hydroperoxides [24]. The sample (0.3 mL) was mixed with the extraction solution (isooctane/isopropanol, *v*/*v*, 3:1; 1.5 mL) and then vortexed for 10 s (three times) with 20 s intervals. The mixture was separated by centrifugation (Cence, Changsha, China) at 2000× *g* for 2 min (4 °C) and the supernatant (200 µL) was added to MB (methanol–butanol, *v*/*v*, 2:1; 2.8 mL). To each sample, Fe^2+^ solution (50 µL) was added with ammonium thiocyanate solution (50 µL, 3.94 M), rapidly vortexed, and then reacted for 20 min at room temperature and protected from light. Sample absorbance was analyzed at 510 nm with a UV/Vis spectrophotometer (Shimadzu, UV-2700, Kyoto, Japan). Fe^2+^ solutions were obtained by centrifugation (2000× *g*, 2 min) of a freshly prepared mixture of FeSO_4_ (1 mL, 0.144 M)) and BaCl_2_ (1 mL, 0.132 M, in 0.4 M HCl). The standard curve was constructed using cumene hydroperoxide (CH).

The TBARS quantification method was somewhat refined according to the description of McDonald and Hultin [25]. The emulsion (1 mL) was mixed with thiobarbituric acid (TBA) reagent (2 mL), which contains 150 g/L trichloroacetic acid (TCA), 3.75 g/L TBA, and 0.25 mol/L HCl. The samples were boiled for 15 min, cooled, mixed with chloroform (1 mL), vortexed, and finally centrifuged (2000× *g*, 15 min). A UV/Vis spectrophotometer was used to determine the absorbance at 532 nm. The lipid concentration was calculated using TBARS with a standard curve of 1,1,3,3-tetraethoxypropane.

### 2.8. Protein Oxidation Analysis

The increase in carbonyl groups and the reduction in free sulfhydryl groups were monitored during emulsion storage. The protein carbonyl content of the samples was determined according to the method provided by Levine et al. [26]. The emulsion was mixed with TCA (200 mg/mL) in equal amounts, incubated in ice water for 10 min, and then centrifuged (4 °C, 15,000× *g*, 10 min). The protein pellet was dissolved in SDS (2 mL, 20 mg/mL, pH 8) solution, and n-hexane (1 mL) was used to separate the oil. A total of 0.1 mL of clear protein solution was taken after centrifugation (2000× *g*, 5 min) and added to DNPH (2 mL, 10 mM) solution and allowed to react for 1 hr. TCA (1 mL, 200 mg/mL) was added, vortexed, and centrifuged (6000× *g*, 10 min) to recover the WPI pellet, and cleaned three times using ethanol/ethyl acetate (1:1, *v*/*v*) solvent. After blowing the residual ethanol/ethyl acetate dry, the WPI pellets were solubilized in guanidine hydrochloride (2 mL, 6.0 M) and incubated at 37 °C for 15 min. Absorbance was collected at 370 nm to analyze the carbonyl content. Results were analyzed using a protein molar extinction coefficient of 22,000 M^−1^cm^−1^.

The content of free sulfhydryl groups in emulsions was determined according to Beveridge’s method [27]. The 100 µL emulsion was thinned to 2 mL with PBS (10 mM, pH 7.0), then blended with 10 mL of Tris-Gly buffer containing urea (8 M, pH 8), and finally 80 µL of Ellman’s reagent (5,5′-dithiobis(2-nitrobenzoic acid) (DNTB)) was added and stirred for 30 min at 25 °C and then centrifuged (Cence, Changsha, China, 12,000× *g*, 10 min). The sulfhydryl content was calculated based on the absorbance at 412 nm and the molar extinction coefficient of 13,600 M^−1^cm^−1^ using PBS as a blank.

### 2.9. Isothermal Titration Calorimetry (ITC)

Raw data was measured at room temperature by a MicroCal PEAQ-ITC (version, 1.41, Malvern Instruments Inc., Northampton, MA, USA), and other thermodynamic parameters (*K_d_*, n, Δ*G*, Δ*H*, and Δ*S*) were calculated and analyzed using MicroCal PEAQ-ITC analysis software [28]. The cuvette was injected with 200 µL of PBS (10 mM) containing WPI (0.3 mM), and the reference cuvette was injected with an equal amount of ultrapure water. The WPI solution was stirred continuously (750 rpm), and 4 µL of sesamol (20 mM) solution was injected precisely into it every 120 s. The unit point combination model was selected for data fitting and calculated according to the Van’t Hoff equation:(2)$$\Delta G=-RT\ln K_{d}$$
(3)ΔG= ΔH−TΔS

### 2.10. Molecular Docking

The non-covalent interactions between sesamol and WPI were analyzed using molecular docking simulation techniques. The most abundant beta-lactoglobulin (β-LG) was selected to represent WPI. The structure of β-LG (PDB ID: 3NPO) was obtained from the Research Collaboratory for Structural Bioinformatics Protein Data Bank (http://www.rcsb.org, accessed on 2 August 2022). The 3D structure of sesamol (Compound CID: 68289) was downloaded from PubChem (https://pubchem.ncbi.nlm.nih.gov/, accessed on 2 August 2022) [29]. Water molecules were removed from the β-LG structure before docking. The β-LG and sesamol were set to rigid and flexible, respectively. AutoDock 4.2 was used to simulate the binding properties of sesamol to β-LG [30]. Finally, the docking result with the lowest energy was selected, visualized, and exported using the Discovery Studio 2020 software (version 4.5.0, Biovea Inc., Omaha, NE, USA).

### 2.11. Interfacial Tension Measurement

The interfacial activity of sesamol was analyzed using a droplet shape analyzer DSA100 (Krüss GmbH, Hamburg, Germany) [21]. The titration module consists of a DS3210 software-controlled single titration device, a disposable syringe with a Luer lock connector (1 mL), and a standard-fit steel needle (NE45, 1.832 mm). A solution of PBS (10 mM, pH 7) with or without WPI (1 wt%) was first drawn into a syringe, and then the needle was inserted into fish oil containing different proportions of sesamol. A precise drop of suspension is extruded by system control at room temperature, capturing the droplet profile every 5 s for 1 h.

### 2.12. Determination of Sesamol Distribution in Emulsions

The distribution of sesamol in the emulsion was determined by the method of Cheng et al. [18]. The sesamol emulsion (4 mL) was centrifuged (4 °C, 10,000× *g*, 1 h), and the aqueous phase was carefully collected with a syringe. PBS (2 mL, 10 mm, pH 7) was added to the remaining emulsified layer, vortexed for 10 min, and centrifuged (4 °C, 3000× *g*, 5 min) to remove the aqueous phase. This was repeated thrice, then 1 mL of isooctane-isopropanol (3:1) was added, vortexed for 10 min, centrifuged (4 °C, 3000× *g*, 5 min) to remove the lower aqueous phase and the intermediate emulsifier layer, and the organic phase (oil phase) was carefully collected using a syringe. The sample (emulsion, oil, or aqueous phase) of 0.2 mL was vortexed in a 5 mL centrifuge tube with 2 mL of methanol (3 min) and then placed in an ultrasonic water bath (5 min). The supernatant was collected, and the process was repeated thrice. The supernatant was transferred to a dark glass vial to evaporate the solvent and fix the volume to 1 mL. The sample (1 mL) was injected into the sample vial with an organic 0.22 µm filter. The sesamol content was analyzed by HPLC LC-20A (Shimadzu Corporation, Kyoto, Japan) and calculated using a standard curve made by dissolving sesamol in methanol. The sesamol distribution ratio in each phase (aqueous phase, oil phase, interfacial layer) is based on the following formula:(4)$$\mathrm{Partitioning}\;\mathrm{ratio}\;\mathrm{in}\;\mathrm{the}\;\mathrm{aqueous}\;\mathrm{phase}\;\left(\%\right)\approx \frac{C_{a}\times V_{a}}{C_{e}\times V_{e}}\times 100\%$$
(5)$$\mathrm{Partitioning}\;\mathrm{ratio}\;\mathrm{in}\;\mathrm{the}\;\mathrm{oil}\;\mathrm{phase}\;\left(\%\right)\approx \frac{C_{o}\times V_{o}}{C_{e}\times V_{e}}\times 100\%$$
(6)$$\mathrm{Partitioning}\;\mathrm{ratio}\;\mathrm{in}\;\mathrm{the}\;\mathrm{interface}\;\mathrm{layer}\;\left(\%\right)\approx 100-\left(4\right)-\left(5\right)$$

*C_e_*, *C_a_*, and *C_o_* are the concentrations of sesamol in the emulsion, aqueous, and oil phases. *V_e_* is the volume of 1 mL of emulsion, *V_a_* and *V_o_* are the volumes of the aqueous phase (0.95 mL) and oil phase (0.05 mL) in 1 mL of emulsion.

### 2.13. Statistical Analysis

All results, obtained via triplicate experiments, are presented as the mean ± standard deviation. Statistical data analysis was performed using SPSS (V20, SPSS Inc., Chicago, IL, USA). Differences between samples were assessed using analysis of variance (ANOVA). *p* < 0.05 was considered statistically significant.

## 3. Results and Discussion

### 3.1. Effect of Sesamol on the Physical Properties of Emulsions

The physical stability of emulsions with and without sesamol was evaluated by monitoring changes in emulsion particle size, charge, and microstructure during storage under the Fenton system for 5 days.

#### 3.1.1. Effect of Sesamol on the Droplet Size of WPI-Stabilized Emulsions during Storage

All the emulsions initially contained relatively small droplets (*D_4,3_* < 1.7 µm; Figure 1A). Nevertheless, the sesamol-added emulsions had lower *D_4,3_* values than the control sample (0.00% sesamol emulsions), and the average droplet size tended to decrease with the increase in sesamol content. The presence of sesamol promoted the production of small oil droplets during the preparation of emulsions. In addition, the oil droplet size of the control emulsion decreased on day 2 of storage and remained at a lower value from day 2 to day 4 (Figure 1A). The larger the oil droplet size, the faster the rate of agglomeration. The newly prepared control emulsion had the largest droplet size and formed a creamy or oil layer on top of the emulsion first because of the flocculation or agglomeration of large droplets. The proportion of small size droplets remaining in the emulsion was much greater, so the light scattering measurement reflects only the size of the small droplets remaining in the emulsion [17]. The addition of 0.09% sesamol not only promoted the formation of small oil droplets during emulsion preparation but also inhibited the degree of droplet aggregation during the first four days of storage. A similar study found that catechins interact with rice bran proteins through hydrogen bonding and hydrophobic interactions and enhanced the ability of WPI as an emulsifier to form and stabilize emulsions [31].

#### 3.1.2. Effect of Sesamol on the *ζ-Potential* of WPI-Stabilized Emulsions during Storage

All the emulsions had a slight negative charge (less than 15 in absolute value; Figure 1B). The pH 7 of the emulsion is higher than the WPI isoelectric point, so the charges of the droplets are all negative [32]. The smaller absolute values of charge may be because ultrasound during emulsion preparation promoted the binding of WPI to polyphenols and reduced the exposure of negatively charged groups on WPI [33]. More sesamol masks more WPI negatively charged groups; thus, the magnitude of the initial ζ-potential depended on the amount of sesamol added: 0.00% > 0.01% > 0.03% > 0.09%. Our observations are consistent with Yi et al., who found that the addition of the antioxidant black rice anthocyanins to the aqueous phase of walnut O/W nanoemulsions resulted in a decrease in the absolute value of the ζ-potential [34]. We observed a decreased trend in the zeta potential of the control emulsion during storage (Figure 1B), indicating that the interfacial composition and structure of the emulsion changed during storage.

#### 3.1.3. Effect of Sesamol on the Visual Appearance and Microstructure of WPI -Stabilized Emulsions during Storage

The visual appearance of the emulsion was recorded after 5 days of storage under the Fenton system (Figure 2A). The freshly prepared emulsion (0 days) was uniformly creamy, indicating that the oil droplets were evenly distributed, free of flocculation and agglomeration, and were not altered by the presence of sesamol. After 5 days of storage, the overall color of the control emulsion gradually turned slightly yellow, and the color change in the top layer was apparent. The products of the Schiff base reaction during the oxidation of the emulsion changed its color to yellow [35], so the color change proved that the formation of the cream layer originated from the oxidation of lipids. Emulsions with 0.01% sesamol were observed to have insignificant color changes, and 0.03 and 0.09% sesamol emulsions maintained a homogeneous milky appearance throughout storage. Meanwhile, the changes in the microstructure of the emulsions were evidently reflected by CLSM observation(Figure 2B). The oil droplets (green) of all emulsions showed aggregation with the increase in storage time, and especially those in the control emulsion showed the most apparent aggregation with the largest oil droplets. As the sesamol content increased, the inhibition of the size and number of aggregated oil droplets and the delay in the appearance of oil droplet aggregation increased. On day 5, all emulsions were observed to have droplet flocculation and coalescence, with the smallest droplets observed for the 0.09% sesamol emulsion as well as the least coalescence.

### 3.2. Effect of Sesamol on the Lipid Oxidation of WPI-Stabilized Emulsions during Storage

The presence of sesamol inhibited the oxidation of lipids in WPI-stabilized emulsions, especially the formation of hydrogen peroxide was reduced by 46.44 (0.01% sesamol), 63.88 (0.03%), and 72.6% (0.09%) after 5 days of storage(Figure 3A). The ability of sesamol to inhibit lipid oxidation was positively correlated with the amount added, with high levels (0.09%) of sesamol having the best antioxidant effect, thereby delaying the accumulation of lipid hydroperoxides in the emulsion by 4 days. The effect of sesamol in inhibiting the accumulation of TBARS in emulsions was similar to that of lipid hydroperoxides. The addition of sesamol significantly reduced the generation of secondary reaction products during storage (*p* < 0.05), further confirming the antioxidant capacity of sesamol(Figure 3B). Similar results have been found with the addition of natural polyphenols to other fish oil-fortified emulsions. For example, lipid oxidation inhibition by resveratrol was observed in fish oil emulsions [36]. The excellent antioxidant properties of sesamol originate from its unique structure and chemical reactivity and the phenolic group on its benzodioxyl group has the same powerful antioxidant scavenging ability as the phenolic hydroxyl group [37]. Sesamol can not only effectively scavenge various oxidative radicals to interrupt the lipid oxidation chain inhibiting oxidation through its powerful electron-donating ability, but can also inhibit the catalyzing lipid oxidation by chelating transition metals [38]. In contrast, the benzodioxole group of sesamol generates another antioxidant (1,2-dihydroxy benzene) when it scavenges hydroxyl radicals; 1,2-dihydroxy benzene can continue to scavenge oxidative free radicals to exert antioxidant effects, and this reaction of sesamol gives it a sustainable antioxidant capacity [10]. In addition, the relationship between emulsifiers and antioxidants on the interfacial layer has a great influence on the stability of lipid oxidation.

### 3.3. Effect of Sesamol on the Protein Oxidation of WPI-Stabilized Emulsions during Storage

For the initial emulsion, the carbonyl content in the sesamol emulsion was lower than that of the control emulsion. The results indicated that sesamol effectively inhibited protein oxidation during the emulsion preparation. Carbonyl groups are generated by protein functional groups through reactions with lipid oxidation products or catalyzed by excess metal ions [39]. The accumulation of carbonyl groups in all emulsions represented a deepening of protein oxidation. However, the degree of protein oxidation in the emulsions decreased significantly with the addition of sesamol (Figure 4A), indicating that sesamol inhibits protein oxidation in emulsions. All the above results were consistent with the lipid oxidation results. Moreover, natural phenols can inhibit the oxidation of proteins in emulsions. For example, tea polyphenols effectively inhibited protein oxidation in whey protein-stabilized emulsions [40]. The protein peptide backbone is attacked by reactive oxygen species to lose hydrogen atoms to form protein radicals, which react with oxygen to form peroxyl radicals, followed by a series of protein oxidation reactions [41]. Sesamol can inhibit protein oxidation through its strong hydrogen supply capacity [38].

Interestingly, the initial sulfhydryl content of the control emulsion was lower than that of the sesamol emulsion(Figure 4B), and this may be due to the fact that sesamol promotes the disruption of protein disulfide bonds (S-S) and the generation of new sulfhydryl groups in the presence of ultrasound [42]. In contrast, sesamol promotes the unfolding of protein structure and the exposure of internal thiol groups [43]. The rate of sulfhydryl loss increased with the amount of sesamol during storage, indicating that the presence of sesamol increased the efficiency of protein sulfhydryl scavenging radicals to inhibit lipid oxidation in emulsions. Previous studies have also shown that protein can replace lipid oxidation as a lipid antioxidant [44]. Although our results suggest that sesamol is an effective antioxidant against lipid and protein oxidation at all the levels tested, the level of sesamol needs to be controlled to avoid negative effects on proteins. At 2 days before emulsion storage, the addition of 0.09% sesamol inhibited the growth trend of lipid hydroperoxides and thiobarbituric acid reactants and only reduced the growth of protein oxidation products. Compared with the control emulsion, the addition of 0.09% sesamol inhibited the formation of 72.6% lipid hydroperoxides and 54.33% thiobarbituric acid reactants in the emulsion after 5 days of storage; however, the reduction of protein sulfhydryl groups and the production of carbonyl groups were little inhibited. It indicates that sesamol mainly inhibits lipid oxidation in emulsions rather than protein oxidation.

### 3.4. Determination of Sesamol Binding to WPI by Isothermal Titration Calorimetry (ITC)

The interaction of sesamol with WPI in the interface is important and related to its antioxidant capacity in the emulsion. Therefore, ITC was used to thermodynamically characterize sesamol and WPI to investigate their non-covalent interactions in this experiment. The heat generated by the interaction of sesamol with WPI was recorded as a function of time, and an integral calculation was performed to obtain the enthalpy curve. The relevant thermodynamic parameters were obtained by model fitting the enthalpy to molar ratio curves (Figure 5). The results show that sesamol bonded spontaneously to WPI and the reaction was exothermic (Δ*G* < 0, Δ*H* < 0). The larger equilibrium dissociation constants (KD) and smaller stoichiometric binding numbers (n) for sesamol and WPI binding were 1.75 × 10^−3^ M and 1.6 × 10^−2^ M, respectively, exhibiting weak affinity and nonspecific binding. The non-covalent interactions between polyphenols and proteins may be driven by hydrophobic, hydrogen bonding, electrostatic interactions, and Van der Waals forces [45]. Since Δ*H* < 0, Δ*s* < 0, and the smaller Gibbs free energy (Δ*G* = −3.76 kcal/mol is closely related to the larger enthalpy and entropy changes, the reaction of sesamol with WPI may mainly be driven by hydrogen bonding and Van der Waals forces [46]. The same results were found in the ITC studies of the ascorbic acid–bovine serum albumin and a-tocopherol–bovine serum albumin systems, where the favorable enthalpy and unfavorable entropy indicated that the main driving forces of the binding reaction were hydrogen bonding forces and Van der Waals forces [47]. The negatively charged O and N atoms on the protein polypeptide chain can form hydrogen bonds with the positively charged hydrogen atoms on the polyphenol phenolic hydroxyl group [48]. In summary, the results indicate that sesamol and WPI interact spontaneously at the interface through non-covalent interactions and aggregate with each other at the interfacial layer to stabilize the emulsion.

### 3.5. Molecular Docking

Molecular docking simulations were used to understand the binding sites and types of interaction forces between WPI and sesamol. Figure 6 shows the best simulation with the lowest binding energy among multiple simulations of sesamol docking with WPI (β-LG) molecules. The 3D docking results show that the binding site of sesamol to β-LG tended to bind to the surface cavity of β-LG rather than the hydrophobic cavity, which may be related to the less hydrophobic nature of sesamol and is consistent with the unfavorable entropic change of ITC. The docking results indicate that the interaction between sesamol and WPI (β-LG) is mediated by not only hydrogen bonding and Van der Waals forces but also hydrophobic interactions. The docking results are consistent with the results of ITC analysis, i.e., hydrogen bonding plays an important role in the binding. As shown in the lower right 2D diagram of Figure 6, there is a hydrogen bonding interaction between Leu-10 of WPI (β-LG) and the methoxy of sesamol, whereas sesamol has Van der Waals forces with five amino acids (Thr-6, Met-7, Lys-8, Gly-9, and Ile-78), hydrophobic and aromatic amino acids (Pro-79 and Ala-80) have hydrophobic interactions (pi-alkyl) with the benzene ring of sesamol. Similar studies have found that hydrogen bonding and hydrophobic interactions are also key interaction forces in the reaction, both between hydroxylated PAHs and peroxidase [29], and between grape skin extract and wheat gliadin [30].

### 3.6. Effect of Sesamol on the Interfacial Activity of WPI Solution with Oil Phase

Generally, the lower the interfacial tension, the higher the aggregation stability of the emulsion [49]. Therefore, we measured the interfacial tension between fish oil and WPI solutions with different concentrations of sesamol at room temperature (Figure 7). WPI adsorption to the oil-water interface changed from fast to slow, and the interfacial tension decreased with the increase in adsorbed WPI at the oil–water interface. This effect resulted from the ability of WPI to adsorb and form interconnected, viscoelastic films at the oil-water interface [50,51]. The addition of sesamol further reduced the interfacial tension, by 19.4% especially at high levels (0.09%). The lower interfacial tension facilitates the formation of small, stable droplets [52]. This is consistent with the previous observation of minimum droplet size and optimal physical stability of high-level sesamol emulsions. The additional reduction in interfacial tension can be attributed to the WPI–sesamol interaction. Hydrogen bonding, Van der Waals forces, and hydrophobic interactions supported and competed with each other to promote the proximity of sesamol to the WPI surface cavity binding site and its aggregation near the binding site [53]. In addition, the further cross-linking of sesamol with the adsorbed layer WPI may form a synergistic mechanism. Dimitris et al. hypothesized that chlorogenic acid at the interface can form hydrogen bonds with multiple adjacent protein molecules, inducing protein unfolding to form a more efficient interfacial coverage [54].

### 3.7. Interfacial Partitioning Analysis of Sesamol in Emulsions

Emulsions containing different concentrations of sesamol showed similar inter-tissue partitioning behavior (Figure 8). The sesamol in the newly prepared emulsions was mainly located in the aqueous phase (54–58%), followed by the interfacial layer (30–33%) and the oil phase (11–12%). In previous studies, sesamol was entirely located in the aqueous phase in freshly prepared emulsions because it was bound to unabsorbed proteins [17]. The partitioning of sesamol in the aqueous phase in this study was lower than in previous studies, which may be related to the treatment with ultrasound during our emulsion preparation. Ultrasound may have promoted the partition of sesamol into the interfacial layer and oil phases.

After 5 days of storage, the total amount of sesamol in emulsion was reduced to one-third of the original level, which was associated with chemical degradation during oxidation [22]. Sesamol content in the oil phase increased significantly during the first 2 days of emulsion storage, indicating that sesamol in the aqueous phase and interfacial layer diffused into the oil phase, increasing the partition ratio of sesamol in the oil phase [55]. After redistribution by diffusion, the partition ratio of the oil phase sesamol reached a maximum, while the partition ratio of the interfacial layer sesamol reached the lowest level. On the third day of storage, the sesamol content in both the oil and aqueous phases decreased by 50%, while the sesamol content in the interfacial layer increased, indicating that some of the sesamol in the oil and aqueous phases diffused into the interfacial layer in addition to oxidative degradation. In summary, the distribution of sesamol in the emulsion during storage is dynamic. We hypothesize that the micelle or vesicle structure in the emulsion facilitates the dynamic diffusion of sesamol between the phases [56].

Accurately determining the interfacial distribution of antioxidants in emulsions is not a simple task. Separating different phases by centrifugation can disrupt the equilibrium of the interfacial region, and the interfacial distribution of antioxidants detected on this basis may not reflect the actual situation. However, there is no method to determine the interfacial distribution of antioxidants by directly measuring the content of antioxidants in each phase. The relatively scientific approach is based on the reaction between the 4-hexadecyl diazenium ion (16-ArN_2_^+^(BF_4_^−^)) molecular probe and the antioxidant, and the kinetic equations are used to calculate the partition constants and interfacial molar values of the antioxidant between different interfaces in the emulsion [57,58,59,60]. In the future, we will validate the accuracy of the centrifugation method using molecular probe methods.

## 4. Conclusions

In conclusion, WPI-stabilized emulsions supplemented with sesamol have superior physical and chemical stability during storage. The presence of sesamol in emulsions has a desirable positive effect, especially in emulsion with a sesamol concentration of 0.09%. ITC and molecular docking results indicated that sesamol is spontaneously bound to WPI mainly through hydrogen bonding, Van der Waals forces, and hydrophobic interactions. In addition, the presence of sesamol reduced the interfacial tension, indicating sesamol-WPI adsorption and interactions at the interface. Sesamol on the interfacial layer exerts antioxidant capacity in a timely and efficient manner and is replenished by diffusion from the oil and aqueous phases when consumed. In summary, sesamol has rich potential for the development of DHA-fortified emulsion delivery systems and the food industry. Our results have important implications for antioxidant studies of PUFA-rich emulsion delivery systems.

## Figures and Tables

**Figure 1 foods-12-01287-f001:**
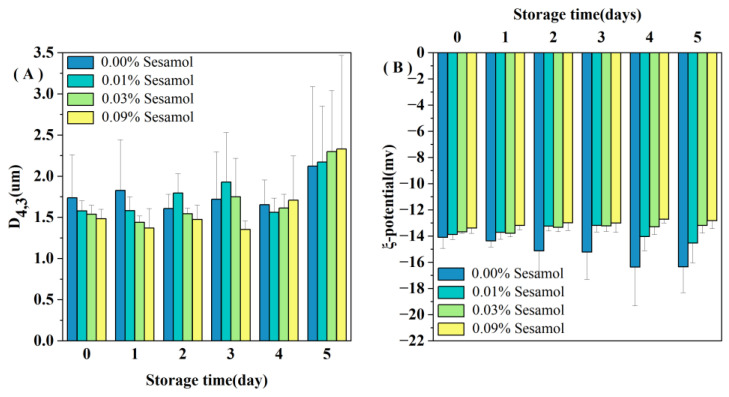
Changes in the mean droplet size D_4,3_ (**A**) and ζ-potential (**B**) of emulsions containing different levels of sesamol during storage (30mL of emulsion in a 50mL glass bottle with screw cap stored at room temperature and protected from light).

**Figure 2 foods-12-01287-f002:**
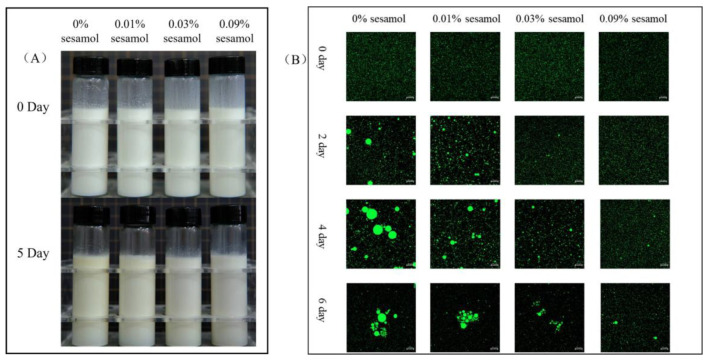
The effects of sesamol on the visual appearance (**A**) (from left to right: 0% sesamol, 0.01% sesamol, 0.03% sesamol and 0.09% sesamol) and emulsions’ confocal micrographs (**B**) (30 mL of emulsion in a 50 mL glass bottle with screw cap stored at room temperature and protected from light).

**Figure 3 foods-12-01287-f003:**
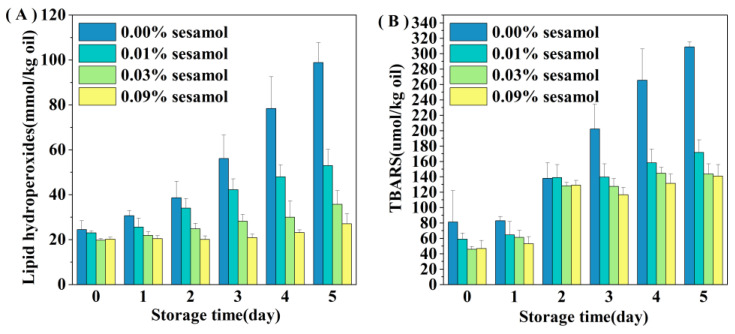
Effect of sesamol on lipid hydrogen peroxide (**A**) and TBARS (**B**) concentrations in WPI-stabilized fish oil emulsions during storage (30 mL of emulsion in a 50 mL glass bottle with screw cap stored at room temperature and protected from light).

**Figure 4 foods-12-01287-f004:**
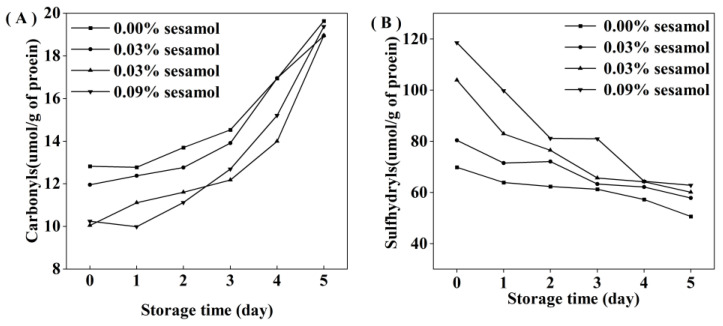
Effect of different sesamol concentrations on protein sulfhydryl (**A**) and carbonyl (**B**) groups in emulsions during storage (30 mL of emulsion in a 50 mL glass bottle with screw cap stored at room temperature and protected from light).

**Figure 5 foods-12-01287-f005:**
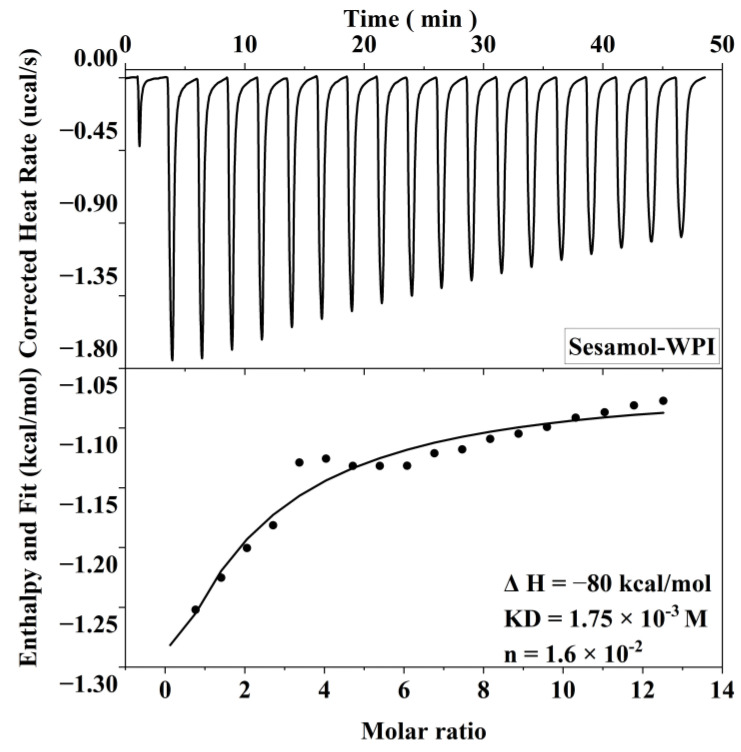
Correction heat rate versus time and enthalpy change corresponding to 20 mM sesamol titration of 0.3 mM WPI.

**Figure 6 foods-12-01287-f006:**
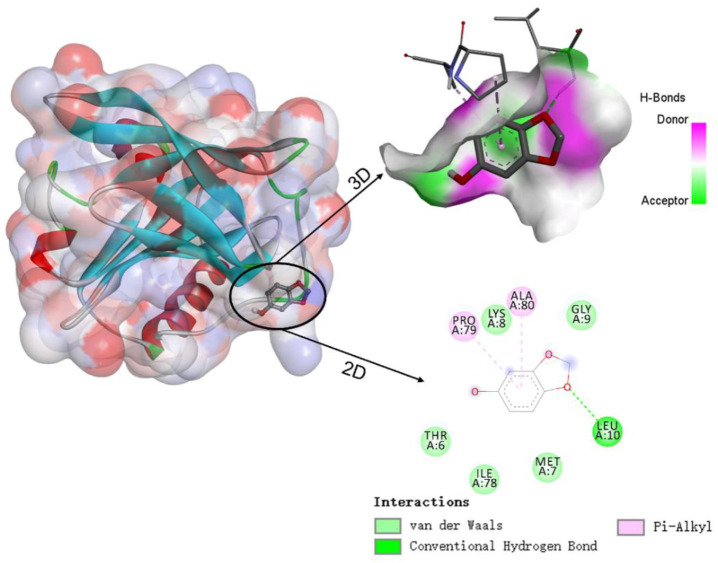
Schematic diagram of the 3D docking model and 2D interaction between β-lactoglobulin (β-LG) and sesamol.

**Figure 7 foods-12-01287-f007:**
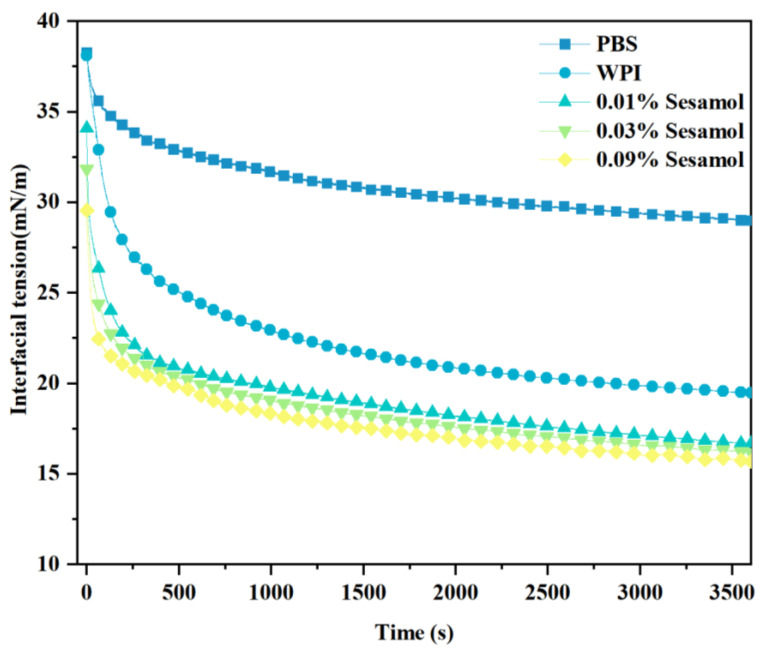
Effect of sesamol on the interfacial tension between the oil phase and WPI solution with time.

**Figure 8 foods-12-01287-f008:**
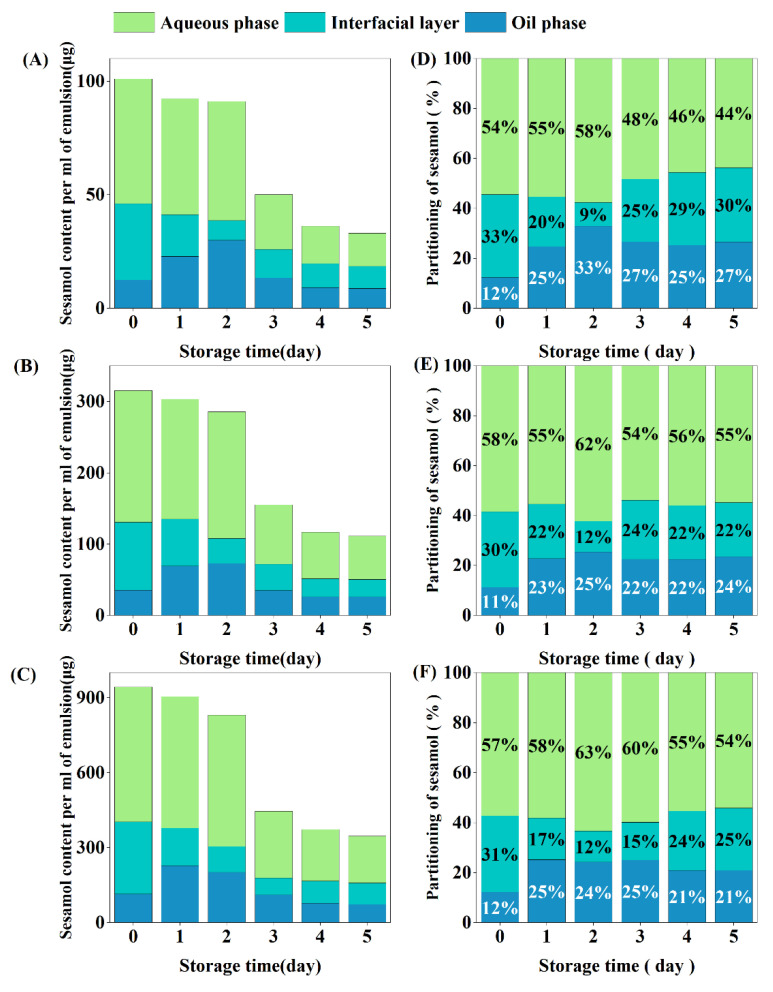
Interfacial partitioning of sesamol in emulsions. Content of sesamol in the aqueous phase, oil phase, and interfacial layer per ml of emulsion ((**A**): 0.01% sesamol, (**B**): 0.03% sesamol, (**C**): 0.09% sesamol). Percentage partitioning of sesamol in the aqueous phase, oil phase and interfacial layer ((**D**): 0.01% sesamol, (**E**): 0.03% sesamol, (**F**): 0.09% sesamol).

## Data Availability

All related data and methods are presented in this paper. Additional inquiries should be addressed to the corresponding author.

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
