# Peer review of "Improvement of Oxidative Stability of Fish Oil-in-Water Emulsions through Partitioning of Sesamol at the Interface"

_foods, 2023, doi:10.3390/foods12061287_

Round 1

Reviewer 1 Report

The paper  “Improvement of oxidative stability of fish oil-in-water emulsions through partitioning of sesamol on the interface” was submitted to Foods for review.

General observations:

In this manuscript, the authors focus on addition of sesamol to prevent the oxidation of polyunsaturated fatty acids in emulsion foods. They focus on the effect of sesamol on the oxidative stability of WPI-stabilized fish oil O/W emulsions with the study of interfacial partitioning and interactions. Numerous techniques (granulometry, zeta potential, microstructure, calorimetry by isothermal titration, etc.) are used to gain an overall understanding of the phenomena.

The English language and style are correct, only minor spelling are required.

The manuscript is of quality, clear, and well documented with many relevant references. However, the techniques used to obtain particle size, zeta potential, and interfacial tension lack important information and clarification is needed on the results obtained. Therefore, for all of the above reasons and my suggestions that follow.

Observations and comments:

Abstract:

Lines 17-18 : “zeta-potential”

Line 18 : “0.09%”. Is it a percentage by weight or other? Please specify.

Materials and methods:

Lines 111-112 p.3 : Please specify the type of %.

Line 115 p.3 : “subsequently sonicated for 5 min at 650 W”. Please specify model and brand of the sonicator. As well as the characteristics of the probe.

Lines 134-135 p.4 : “intensity weighted mean hydrodynamic diameter (Z-average)”. Please add a reference which explains how Z-average is obtained.

2.5 The authors need to add more information about the measurement of zeta potential (cell, Hückel or Helmholtz model, ionic strength of continuous phase, pH).

Line 151 p.4 : “Briefly.” Please correct the sentence.

Lines 158-159 p.4 : “The absorbance was measured at 510 nm using a  UV/Vis spectrophotometer”. Can you justify the value of 510 nm? It would have been better for the understanding to add the UV/Vis spectrum. This can be done in supplementary information.

Lines 159 and 170 p.4 : Choose between UV/Vis and UV-Vis.

Line 231 p.6 : “1.832 mm outer diameter”. Please add the gauge number.

Lines 228 to 238 p.6 : What is the volume of drop? What are the densities of the phases? Did you work with a Bond number over 0.1?

Results and Analysis

3.1.1 In this section, the authors give D(4,3) (also called volume moment mean) of the emulsions while they mention “mean intensity weighted diameter” in p.4. Can the authors clarify this point? Furthermore, the authors have reported only mean whereas it is strongly recommended to always report at least one size distribution to get the emulsion profile. Please add in annex or supplementary information the PSD (particle size distribution).

The different sizes of emulsions reported are between 1.5 and 2.5 microns (see figure 2). It is specified in the material and method section that the emulsion sizes were measured by DLS. DLS is a technique that works well for very small sizes, between 10 nm and 1000 nm. The sizes obtained are in the high range of the device, and often the results obtained are not repeatable and not trustworthy. Can the authors indicate which polydispersity index values are obtained? I am also very surprised about the values of standard deviation, are they calculated from D(4,3) or Z-average values?

Line 295 p.8 : “decreased significantly”. Can you comment on the term “significantly” because it is not obvious to me?

Lines 297 to 300 p.8 : I suggest to comment more because it is not clear.

Line 325 p.8: “We observed a significant increase in the negative charge on the lipid droplets”. The difference between 0 and 5 days of storage is never more than 2 mV and is within the error bar, so in my opinion it is difficult to conclude that there is a significant increase in the negative charge. Did the authors compare the electrophoretic mobility to the zeta potential in order to be sure of the values obtained, knowing that the electrophoretic mobility is the quantity directly measured by the device?

Lines 339-340 p.9 : I suggest to comment more.

Line 446 p.12 : “1.75 × 10-3 M and 1.6 × 10-2 M”

Lines 499 to 501 p.14 : “The adsorbance of WPI to the oil-water interface decreased as time increased, and the tension at the oil-water interface decreased.” What do the authors mean by “adsorbance”, maybe adsorption? The authors need to rephrase because this sentence seems to imply that the interfacial tension decreases with time because there is less WPI at the interface with time.

Lines 501 to 503 p.14 : “This effect results from the ability of WPI to adsorb and form interconnected, viscoelastic thick films at the oil-water interface.” The authors need to add at least 2 references to justify the viscoelastic behavior of the interface.

Line 505 p.14 : Two times reference [41].

Figure 8 p.15 : I suggest adding color to better differentiate the curves.

Author Response

Dear reviewer 1,

Thank you for your useful comments and suggestions on our manuscript. We have modified the manuscript accordingly, and detailed corrections are listed below point by point:

Abstract:

Point 1:Lines 17-18 : “zeta-potential”.

Response: Thanks for your suggestions. It has been changed to "zeta-potential" in line 18.

Point 2: Line 18 : “0.09%”. Is it a percentage by weight or other? Please specify.

Response: % is the mass to volume ratio. Notes have been added to the text in line 19.

Materials and methods:

Point 3: Lines 111-112 p.3 : Please specify the type of %.

Response: % is the mass to volume ratio. Notes have been added to the text in line 109.

Point 4: Line 115 p.3 : “subsequently sonicated for 5 min at 650 W”. Please specify model and brand of the sonicator. As well as the characteristics of the probe.

Response 4: Information about the model and brand of the sonicator, as well as the model of the detector, has been added in line 112:The 5% (v/v) oil phase and 95% (v/v) water phase were mixed and processed by a high-speed stirring device (IKA, T25, Germany) at 15,000 rpm for 2 min, followed by placing the primary emulsion on an ice bath and subjecting it to 650 W ultrasonication (Nanjing Xianou Instrument Manufacturing Co., Ltd., XO-1000D, China, amplifying bar Φ 6) for 5 min.

Point 5: Lines 134-135 p.4 : “intensity weighted mean hydrodynamic diameter (Z-average)”. Please add a reference which explains how Z-average is obtained.

Response 5: Calculation equations and references have been added in line 121-123:

The results were recorded as volume-weighted average particle size (D4 , 3)

                  (1)

ni: number of droplets, di: droplet diameter.

Point 6: 2.5 The authors need to add more information about the measurement of zeta potential (cell, Hückel or Helmholtz model, ionic strength of continuous phase, pH).

Response 6: The relevant information has been added in line 125-128:The zeta potential was measured to use dynamic light scattering (Malvern, NANO ZS90, UK). The emulsion samples were diluted 200 times with PBS (10 mM, pH 7) at 25°C and loaded into DTS1070 measuring dishes. The Hückel model is selected for the measurement calculation .

Point 7: Line 151 p.4 : “Briefly.” Please correct the sentence.

Response 7: The sentence has been corrected in line 139.

Point 8: Lines 158-159 p.4 : “The absorbance was measured at 510 nm using a  UV/Vis spectrophotometer”. Can you justify the value of 510 nm? It would have been better for the understanding to add the UV/Vis spectrum. This can be done in supplementary information.

Response 8: we scanned the absorbance of lipid peroxides from 400nm to 600nm (Figure 1). The results showed that the maximum absorbance wavelength is 510nm. At the same time, the literature from two papers also support this. So the absorbance values were measured using 510 nm.

Zhu, Z.; Zhao, C.; Yi, J.; Liu, N.; Cao, Y.; Decker, E.A.; McClements, D.J. Impact of Interfacial Composition on Lipid and Protein Co-Oxidation in Oil-in-Water Emulsions Containing Mixed Emulisifers. J Agric Food Chem 2018, 66, 4458-4468, doi:10.1021/acs.jafc.8b00590.

Cheng, C.; Yu, X.; McClements, D.J.; Huang, Q.; Tang, H.; Yu, K.; Xiang, X.; Chen, P.; Wang, X.; Deng, Q. Effect of flaxseed polyphenols on physical stability and oxidative stability of flaxseed oil-in-water nanoemulsions. Food Chem 2019, 301, 125207, doi:10.1016/j.foodchem.2019.125207.

 Appendix Figure 1. Wavelength scan spectrum for measuring lipid peroxides

Point 9: Lines 159 and 170 p.4 : Choose between UV/Vis and UV-Vis.

Response 9: UV-Vis has been changed to UV/Vis in line 153.

Point 10: Line 231 p.6 : “1.832 mm outer diameter”. Please add the gauge number.

Response 10: Needle model is NE45, have been added in line 203:The interfacial activity of sesamol was analyzed using a droplet shape analyzer DSA100 (Krüss GmbH, Hamburg, Germany). The titration module consists of a DS3210 software-controlled single titration device, a disposable syringe with Lu-er-Lock connector (1 ml) and a standard-fit steel needle (NE45, 1.832 mm).

Results and Analysis

Point 11: 3.1.1 In this section, the authors give D(4,3) (also called volume moment mean) of the emulsions while they mention “mean intensity weighted diameter” in p.4. Can the authors clarify this point? Furthermore, the authors have reported only mean whereas it is strongly recommended to always report at least one size distribution to get the emulsion profile. Please add in annex or supplementary information the PSD (particle size distribution).

The different sizes of emulsions reported are between 1.5 and 2.5 microns (see figure 2). It is specified in the material and method section that the emulsion sizes were measured by DLS. DLS is a technique that works well for very small sizes, between 10 nm and 1000 nm. The sizes obtained are in the high range of the device, and often the results obtained are not repeatable and not trustworthy. Can the authors indicate which polydispersity index values are obtained? I am also very surprised about the values of standard deviation, are they calculated from D(4,3) or Z-average values?

Response 11:

(1). Due to my oversight, I wrote D4,3 in the method incorrectly as the average intensity weighted diameter. I apologize and have corrected it to:The results were recorded as volume-weighted average particle size D4 , 3 in line 122:

(2).PSD (Particle Size Distribution) has been added in the attachment.

Appendix Figure 2. The particle size PSD profiles of 0.00% sesamol emulsion, 0.01% sesamol emulsion, 0.03% sesamol emulsion, and 0.09% sesamol emulsion were A, B, C, and D on day 0, and E, F, G, and H on day 5.

  • I apologize about writing the wrong instrument for measuring particle size.The methodology section has been modified in line 120-122:The laser particle size analyzer (Microtrac MRB, S3500, USA) measured the parti-cle size of the emulsion with refractive indices set to 1.49 (oil) and 1.33 (deionized wa-ter).
  • . We are sorry that no polydispersity index values were obtained from the laser particle sizer.

(5). The standard deviation is calculated according to D(4, 3). I apologize for previously using three measurements of the same sample to calculate the standard deviation.This has now been corrected by calculating the standard deviation from three parallel samples. The recreated figure is as follows.

Point 12:Line 295 p.8 : “decreased significantly”. Can you comment on the term “significantly” because it is not obvious to me?

Response 12:Very sorry to have used an inappropriate word. It has been corrected in line 257-259. In addition, the oil droplet size of the control emulsion decreased on 2-day storage and remained at a lower value from  2 day to 4 day (Figure 1A).

Point 13:Lines 297 to 300 p.8 : I suggest to comment more because it is not clear.

Response 13: It has been re-explained in line 254-259:The larger the oil droplet size, the faster the rate of agglomeration. The newly prepared control emulsion has the largest droplet size and it will form a creamy layer or oil layer on top of the emulsion first because of flocculation or agglomeration of large size droplets. The proportion of small size droplets remaining in the emulsion is much larger, so the light scattering measurement reflects only the size of the small droplets remaining in the emulsion.

Point 14: Line 325 p.8: “We observed a significant increase in the negative charge on the lipid droplets”. The difference between 0 and 5 days of storage is never more than 2 mV and is within the error bar, so in my opinion it is difficult to conclude that there is a significant increase in the negative charge. Did the authors compare the electrophoretic mobility to the zeta potential in order to be sure of the values obtained, knowing that the electrophoretic mobility is the quantity directly measured by the device?

Response 14:

  • I agree with your view.I previously focused on the significance analysis results but ignored:its accuracy is on the order of ±10% or ±2mV, whichever is greater.Modify to in line 275-278:We observed a decreased trend in the zeta potential of the control emulsion during storage (Figure 1B), indicating that the interfacial composition and structure of the emulsion changed during storage.
  • I'm sorry I didn't compare electrophoretic mobility and zeta potential.But I learned that the zeta potential (ζ) is obtained indirectly by measuring the electrophoretic mobility (particle velocity divided by electric field strength) under an applied electric field and by calculating it through the Henry equation.

Point 15:Lines 339-340 p.9 : I suggest to comment more.

Response 15: It has been corrected in line 286-288. The products of the Schiff base reaction during the oxidation of the emulsion change its color to yellow[33], so the color change proves that the formation of the cream layer originates from the oxidation of lipids.

Point 16:Line 446 p.12 : “1.75 × 10-3 M and 1.6 × 10-2 M”

Response 16: Correction has been made in line 273.

Point 17:Lines 499 to 501 p.14 : “The adsorbance of WPI to the oil-water interface decreased as time increased, and the tension at the oil-water interface decreased.” What do the authors mean by “adsorbance”, maybe adsorption? The authors need to rephrase because this sentence seems to imply that the interfacial tension decreases with time because there is less WPI at the interface with time.

Response 17: It has been rephrased in line 417-421:WPI adsorption to the oil-water interface from fast to slow, and the interfacial tension decreases with the increase in the number of adsorbed WPI at the oil-water interface. This effect results from the ability of WPI to adsorb and form interconnected, viscoelastic films at the oil-water interface

Point 18:Lines 501 to 503 p.14 : “This effect results from the ability of WPI to adsorb and form interconnected, viscoelastic thick films at the oil-water interface.” The authors need to add at least 2 references to justify the viscoelastic behavior of the interface.

Response 18: They have been added in line 421:

49.Richter, M.J.; Schulz, A.; Subkowski, T.; Boker, A. Adsorption and rheological behavior of an amphiphilic protein at oil/water interfaces. J COLLOID INTERF SCI 2016, 479, 199-206, doi:10.1016/j.jcis.2016.06.062.

50.Schröder, A.; Berton-Carabin, C.; Venema, P.; Cornacchia, L. Interfacial properties of whey protein and whey protein hydrolysates and their influence on O/W emulsion stability. FOOD HYDROCOLLOID 2017, 73. 129-140. doi:10.1016/j.foodhyd.2017.06.001.

Point 19:Line 505 p.14 : Two times reference [41].

Response 19: Redundant has been removed line 423.

Point 20:Figure 8 p.15 : I suggest adding color to better differentiate the curves.

Response 20: They have been changed to different colors in Figure 7. Effect of sesamol on the interfacial tension between oil phase and WPI solution with time.

Reviewer 2 Report

The authors present a very interesting work on the role of sesamol in the oxidative stability of fish oil-in-water emulsions.

In general, the manuscript is well written but the authors must pay attention to verb tenses, sometimes are not used properly.

From the scientific part, the manuscript is very interesting but I have a few comments that the authors should address.

1) The word "migration" used in the abstract and other parts of the text is not appropriate as it indicates that sesamol is first confined in a particular region of the emulsion and then is displaced from there to the interfacial region.

If the authors prepared the emulsion propertly, as it seems they did, then ALL components should be distributed withing the emulsion accoding to the solubilities in each region, and a dynamic equilibrium should exist so that at any time a sesamol molecule reacts it is immediately replaced by another one so that the equilibrium condition still holds.

Other wise, the authors should prove that transport of sesamol from one region to other is rate-limiting.

2) The rupture of the emulsion to determine the distribution of sesamol is not appropriate because analyses is donde AFTER breaking down the emulsion, and so analytical results are biased as they are totally different from those in the intact emulsion.

3) Sesamol is only slightly soluble in water (38 g/L), with a logP value around 2. It means that the concentration in the oil is about 100 times that in water. So, how comes that the authors find that there is more sesamol in water (54-58%) than the sum of that in the interfacial and oil regions ? Maybe this has something to do with the comment in point 2 raised before?

4) A las comment. How the protein oxidation rate compares with the lipid oxidation rate? The authors should indicate it as from their results one might think that protein oxidation is faster than lipid oxidation so that sesamol mainly prevents protein oxidation and not fish oxidation (which must be quite fast because of the hughe number of unsaturations in EPA and DHA and other components)-

Author Response

Dear reviewer 2,

Thank you for your useful comments and suggestions on our manuscript. We have modified the manuscript accordingly, and detailed corrections are listed below point by point:

Point 1: The word "migration" used in the abstract and other parts of the text is not appropriate as it indicates that sesamol is first confined in a particular region of the emulsion and then is displaced from there to the interfacial region.

If the authors prepared the emulsion propertly, as it seems they did, then ALL components should be distributed withing the emulsion accoding to the solubilities in each region, and a dynamic equilibrium should exist so that at any time a sesamol molecule reacts it is immediately replaced by another one so that the equilibrium condition still holds.

Other wise, the authors should prove that transport of sesamol from one region to other is rate-limiting.

Response 1: First of all, thank you for your suggestion, and secondly, we agreed with you after looking for information and discussing.We thought that sesamol should diffuse rapidly between phases, so we changed the word to "diffusion" in line 26、452、459.

Point 2: The rupture of the emulsion to determine the distribution of sesamol is not appropriate because analyses is donde AFTER breaking down the emulsion, and so analytical results are biased as they are totally different from those in the intact emulsion.

Response 2: Thank you for your suggestion. I agree with you that the results of determining the sesamol distribution in the emulsion by centrifugation are biased. However, this is the common method currently used in the literature to measure the distribution of phenolics in emulsions. We also carried out some experiment on the sesamol content in different region of emulsion. The results of sesamol content could show the trend variation of sesamol in emulsion. Moreover, there are some literatures on this and we also talked about this with the authors. In the next step, we will cross-corroborate the sesamol distribution based on a chemical probe 4-hexadecylbenzenediazonium ion (16-ArN2+), kinetic analysis and centrifugation.

Literatures:

Cheng, C.; Yu, X.; McClements, D.J.; Huang, Q.; Tang, H.; Yu, K.; Xiang, X.; Chen, P.; Wang, X.; Deng, Q. Effect of flaxseed polyphenols on physical stability and oxidative stability of flaxseed oil-in-water nanoemulsions. Food Chem 2019, 301, 125207, doi: 10.1016/j.foodchem.2019.125207.

Wang, X.; Yu, K.; Cheng, C.; Peng, D.; Yu, X.; Chen, H.; Chen, Y.; Julian McClements, D.; Deng, Q. Effect of sesamol on the physical and chemical stability of plant-based flaxseed oil-in-water emulsions stabilized by proteins or phospholipids. Food Funct 2021, 12, 2090-2101, doi:10.1039/d0fo02420a.

Point 3: Sesamol is only slightly soluble in water (38 g/L), with a logP value around 2. It means that the concentration in the oil is about 100 times that in water. So, how comes that the authors find that there is more sesamol in water (54-58%) than the sum of that in the interfacial and oil regions? Maybe this has something to do with the comment in point 2 raised before?

Response 3:The maximum added concentration of sesamol in the emulsion was 0.9 g/L, which is much smaller than the solubility in water (38 g/L).The freshly prepared 0.01% sesamol emulsion, 0.03% sesamol emulsion, and 0.09% sesamol emulsion had concentrations of 250.48, 709.47, and 2296.94 μg/ml in the oil phase, which were much higher than those of 57.87, 193.86, and 567.96 μg/ml in the aqueous phase. I apologize for the lack of detail in my description of the information on sesamol distribution ratio in the method. Has changed to:

Sesamol distribution ratio in each phase (aqueous phase, oil phase, interfacial layer) is based on the following formula:

          (2)

            (3)

           (4)

Ce, Ca, and Co are the concentrations of sesamol in the emulsion, aqueous, and oil phases. Ve is the volume of 1 ml of emulsion, Va and Vo are the volumes of aqueous phase (0.95 ml) and oil phase (0.05 ml) in 1 ml of emulsion.

Point4: A las comment. How the protein oxidation rate compares with the lipid oxidation rate? The authors should indicate it as from their results one might think that protein oxidation is faster than lipid oxidation so that sesamol mainly prevents protein oxidation and not fish oxidation (which must be quite fast because of the hughe number of unsaturations in EPA and DHA and other components)-

Response 4: Has been added in line 356-363: At 2 days prior to emulsion storage, the addition of 0.09% sesamol inhibited the growth trend of lipid hydroperoxides and thiobarbituric acid reactants and only reduced the amount of growth of protein oxidation products. Compared to the control emulsion, the addition of 0.09% sesamol inhibited the formation of 72.6% lipid hydroperoxide and 54.33% thiobarbituric acid reactants in the emulsion after 5 days of storage, how-ever, the reduction of protein sulfhydryl groups and the production of carbonyl groups were largely not inhibited. It indicates that sesamol mainly inhibits lipid oxidation in emulsions rather than protein oxidation.

Reviewer 3 Report

The study presents the results of the effect of sesamol concentration on the physicochemical stability of whey protein isolate-stabilized fish oil emulsions. The above as an alternative for its application in the food industry, cosmetics, among others. I consider that the study shows scientific rigor, is novel and of interest for its future application in functional foods. The methodology is clear, concise and I consider that it´s reproducible. The results are adequately described and compared with similar studies. I like the way you presented the tables, graphs, and figures. Furthermore, the conclusions are limited to the relevant findings of the experiment performed.

The suggestion to improve the manuscript is the following: in references section, verify that the name of the journal is abbreviated, as specified in the author´s guide.

Author Response

Dear reviewer 3,

Thank you for your useful comments and suggestions on our manuscript. We have modified the manuscript accordingly, and detailed corrections are listed below.

  1. The suggestion to improve the manuscript is the following: in references section, verify that the name of the journal is abbreviated, as specified in the author´s guide.

Response: Thanks for your suggestions. They have been changed and verified.

Reviewer 4 Report

Review of foods-2232078

This is interesting and good work. However, the authors have used way too much speculation on their data without much proof. Please see my comments below, which must be addressed.

One very important point: Section 2.12 and line 265: when you get the cream layer after centrifuge, it should have the interfacial layer. It is unclear how you could separate the interface from the oil droplets. How is that possible? So the equation about calculating the interfacial portioning does not seem right. This must be properly addressed or the expt re-done before this data can be used in the paper.

There is no ref to the molecular docking work. It must be provided and later, the results from the ref should also be discussed to support your work.

The authors use a range of font size in the manuscript. Not sure why. It must be fixed.

Abstract:

The abstract needs work. It is not clear enough for someone to understand your work. 

Specify to which phase you added sesamol. 

You should not just write “focusing on the interactions and interfacial partitioning of sesamol at the emulsion interface” – it does not make it clear. 

L 18: 0.09% is not a high level of sesamol. 

L 20: Sesamol does not reduce hydrogen peroxide. It should be lipid peroxide radicals or simply lipid oxidation

L24: what do you mean by pro-oxidant molecule? Did you measure which are these? Don’t write something that you have not actually determined.

L 26: how did you know tighter and thicker barrier? Did you show proof? Don’t write in the abstract if it is not experimentally shown.

L 26: so just inhibiting the peroxyl radicals? Nothing else?

Introduction:

L36: I don’t think this is right. Humans can eat enough fish to get the PUFAs you are talking about. Not everyone takes fish oil supplements and they are just fine. 

L57: do yo have ref for using Sesamol as medicine? If not, remove it.

L59: Fix grammar. Past tense

L62-64: give data and more details.

L72: ref 16 is very similar work as your. Please give details of that work, and discuss how yours differs from that work.

L81: remove the word “regularities”

Why did you strip the fish oil? Please give a rationale.

What temp was the emulsion made during processing? It must be clearly written. Any temp abuse here would change your work.

L116: to which phase was sodium azide added?

Fig 1 is not needed at all. Just write the process clearly. That should be enough.

L121: what is a Fenton system? Give full details with ref.

L 191: give ref #

Give ref for ITC work.

L243: This should not be oil. It should cream layer

:243-4: If you centrifuge 0.2 ml emulsion, how much cream and aq phase did you collect? Is that even possible? That seems too less.

L245: 2 ml methanol was added to which phase and to much how much of that phase. Please clarify in the manuscript.

L273: why results and analysis? Why no results and discussion?

What os the pH of the emulsion? Please provide this data and use it to explain your results.

L278: here and all similar cases: Particle size should be replaced with droplet size

L294: clarify what is your control emulsion

L294: clarify what is the control emulsion?

L 296: it did not remain a lower value after 5 days

L296-297: How did you know aggregation and also deoiling? It is not possible from your data here.

L302: It did not inhibit on day 5. Please fix

L307:What data showed that repulsive forces between the droplets increased? You do not have any proof. Please fix.

L 320: yOU should show this by using just WPI and polyphenol without the emulsion

L 323-4: did they add the antioxidant to the emulsion? Or to oil or to the water? Please clarify

L338: How did you know deoiling. There is not clear proof. It might be just creaming.

L340: how come oxidation leads to deoiling? It Is not clear at all.

The visual observation images are not entirely clear to understand any phase seperation or the claimed deoiling.

L352-355: This is not clear from the figure. Better explanation and proof needed.

L355: this result does not prove lipid oxidation

The first paragraph of Section 3.2 and similar sections: These first paragraphs are just repetitions of what you said in methods and are not needed. I suggest that you remove these from the manuscript. It would be better for reading.

Fig 4: what is the x-axis here?

Fig 4 and similar fig caption: Need to give full details in the caption. What temp? what condition, the vials were stored etc.

Section 3.3: mechanism of protein oxidation should be discussed in more details.

L444-6: It is not clear how you calculated the KD and other values from the data. Give more details and ref.

L449: You did not show how you got s<0

L457-9: The explanation is not clear.

Section 3.5: the first sentence is not needed.

L502: The WPi films are not very thick. Please be cautious what you write.

L510: not clear what you mean by aggregation of Sesamol. Please explain better.

L514: this interaction with multiple proteins theory has no proof. Please discuss it with more details with solid evidence.

L 517: your data does not support this claim. Again, you cant say this type of speculation withoit actual data.

L519-521: Did this ref actually show this with proof? How did you they do it> please give more details and explain better.

Section 3.7: this first paragraph is not needed

L 538: are you talking about one or multiple studies here?

L554: not clear how you can say the tendency to migrate to the oil phase, when the Sesamol was added to the oil phase at the beginning. Did you entirely move to aq phase and then migrate back> Is that even possible? Also, see my comments on your wrong calculation of partitioning at the beginning. This must be redone.

Your explanation in 557-567 is not clear. This must be rewritten based on better data.
L569-571: You can't say that. You only studied for 5 days. You don’t know what will happen after that time.

L578-580: This is not clear from the figure. Please fix.

L583-586: there is no proof of this statement.

Author Response

Dear reviewer 4,

Thank you for your useful comments and suggestions on our manuscript. We have modified the manuscript accordingly, and detailed corrections are listed below point by point:

Point 1: One very important point: Section 2.12 and line 265: when you get the cream layer after centrifuge, it should have the interfacial layer. It is unclear how you could separate the interface from the oil droplets. How is that possible? So the equation about calculating the interfacial portioning does not seem right. This must be properly addressed or the expt re-done before this data can be used in the paper.

Response 1: Sorry for not detailing the removal of the interface layer, details have been added in line 211-216:the sesamol emulsion (4 mL) was centrifuged (4°C, 10,000 g, 1 h) and the aqueous phase was carefully collected a syringe. Added 2 ml of PBS (10 mm, pH 7) to the remained emulsified layer, vortexed for 10 min, centrifuged (4°C, 3000 g, 5 min) to remove the aqueous phase, and repeated 3 times, then added 1 ml of isooc-tane-isopropanol (3:1), vortexed for 10 min and centrifuged (4°C, 3000 g, 5 min) to remove the lower aqueous phase and the intermediate emulsifier layer, and carefully collected the organic phase(oil phase) using a syringe .

Sesamol distribution ratio in each phase (aqueous phase, oil phase, interfacial layer) is based on the following formula:

          (4)

            (5)

Ce, Ca, and Co are the concentrations of sesamol in the emulsion, aqueous, and oil phases. Ve is the volume of 1 ml of emulsion, Va and Vo are the volumes of aqueous phase (0.95 ml) and oil phase (0.05 ml) in 1 ml of emulsion.

Point 2: There is no ref to the molecular docking work. It must be provided and later, the results from the ref should also be discussed to support your work.

Response 2: Two references (27 and 28) have been added:

  1. Zhang, J.; Chen, L.; Zhu, Y.; Zhang, Y. Study on the molecular interactions of hydroxylated polycyclic aromatic hydrocarbons with catalase using multi-spectral methods combined with molecular docking. Food Chem 2020, 309, 125743, doi:10.1016/j.foodchem.2019.125743.
  2. Guo, Z.; Huang, Y.; Huang, J.; Li, S.; Zhu, Z.; Deng, Q.; Cheng, S. Formation of protein-anthocyanin complex induced by grape skin extracts interacting with wheat gliadins: Multi-spectroscopy and molecular docking analysis. Food Chem 2022, 385, 132702, doi:10.1016/j.foodchem.2022.132702.

Related discussions have been added in line 409-411:Similar studies have found that hydrogen bonding and hydrophobic interactions are also key interaction forces in the reaction, both between grape skin extract and wheat gliadin, and between hydroxylated PAHs and peroxidase.

Point 3: The authors use a range of font size in the manuscript. Not sure why. It must be fixed.

Response 3:They have been modified to agree with the font size.

Abstract:

Point 4: The abstract needs work. It is not clear enough for someone to understand your work. 

Response 4:It has been corrected in line 13-27:The susceptibility of polyunsaturated fatty acids to oxidation severely limits their application in functional emulsified foods. In this study, sesamol was added to fish oil to make emulsions to investigate the effect of its concentration on the physicochemical properties of WPI-stabilized fish oil emulsions, focusing on the interaction, interfacial activity and interfacial partitioning behavior between sesamol and WPI. The results relating to particle size, zeta-potential, microstructure, and appearance showed that high levels of sesamol (0.09%,w/w) promoted the formation of small oil droplets and inhibited oil droplet aggregation. Furthermore, the addition of sesamol significantly reduced the formation of hydrogen peroxide, generation of secondary reaction products during storage, and degree of protein oxidation in the emulsions. Molecular docking and isothermal titration calorimetry showed that the interaction between sesamol and β-LG was mainly mediated by hydrogen bonds, van der Waals forces and hydro-phobic interactions. Our results show that the interaction of sesamol with WPI reduces the interfacial tension and improves the physical stability of the emulsion, and the dynamic diffusion of sesamol at the interface ensures the oxidative stability of the emulsion.

Point 5:Specify to which phase you added sesamol. 

Response 5:It has been corrected in line 14-15:In this study, sesamol was added to fish oil to make emulsions to investigate the effect of its concentration on the physicochemical properties of WPI-stabilized fish oil emulsions.

Point 6: You should not just write “focusing on the interactions and interfacial partitioning of sesamol at the emulsion interface” – it does not make it clear. 、

Response 6: It has been corrected in line 17:The focus is on the relationship between Sesamol-WPI interactions and interfacial behavior.

Point 7: L 18: 0.09% is not a high level of sesamol. 

Response 7: It has been corrected to in line 18:0.09% (w/v) sesamol

Point 8: L 20: Sesamol does not reduce hydrogen peroxide. It should be lipid peroxide radicals or simply lipid oxidation

Response 8: It has been deleted.

Point 9: L24: what do you mean by pro-oxidant molecule? Did you measure which are these? Don’t write something that you have not actually determined.

Response 9: It has been deleted.

Point 10: L 26: how did you know tighter and thicker barrier? Did you show proof? Don’t write in the abstract if it is not experimentally shown.

Response 10: It has been deleted.

Point 11:L 26: so just inhibiting the peroxyl radicals? Nothing else?

Response 11: It has been corrected in line 24-26: Our results show that sesamol binds to interfacial proteins mainly through hydrogen bonding, and increasing the interfacial sesamol content reduces the interfacial tension and improves the physical and oxidative stability of the emulsion.

Introduction:

Point 12: L36: I don’t think this is right. Humans can eat enough fish to get the PUFAs you are talking about. Not everyone takes fish oil supplements and they are just fine. 

Response 12: It has been deleted.

Point 13: L57: do yo have ref for using Sesamol as medicine? If not, remove it.

Response 13: It has been deleted.

Point 14: L59: Fix grammar. Past tense

Response 14: Has been corrected in line 56: For example, sesamol significantly improved the oxidative stability of lipids in beeswax organogel systems .

Point 15: L62-64: give data and more details.

Response 15: Specific information has been added in line 58-61: The DPPH radical scavenging rate of the same concentration of sesamol (2.5 umol/g) was 2.56 times higher than that of 2,6-di-tert-butyl-p-cresol in lard after thermal in-duction at 180 °C for 80 min.

Point 16: L72: ref 16 is very similar work as your. Please give details of that

Response 16: Specific information has been added in line 66-70 : Wang et al. found that sesamol effectively inhibited particle aggregation and lipid oxidation in protein-stabilized flaxseed oil-in-water emulsions, and hypothesized that sesamol molecules could adsorb on the surface of oil droplets and interact with emulsifiers to influence interfacial properties, thereby enhancing the stability of emulsions

Point 17: work, and discuss how yours differs from that work.

Response 17: Specific information has been added in line 70-75:However, the interaction between sesamol and emulsifiers was not explored and the partitioning of sesamol in all phases of the emulsion was not clarified. Therefore, we expected to understand the relationship between antioxi-dant-emulsifier interactions and interfacial partitioning and the effect on emulsion stability by further investigat-ing the interaction and interfacial partitioning of sesamol in WPI-stabilized fish oil emulsions.

Point 18: L81: remove the word “regularities”

Response 18: It has been deleted.

Point 19: Why did you strip the fish oil? Please give a rationale.

What temp was the emulsion made during processing? It must be clearly written. Any temp abuse here would change your work.

Response 19: Reason and temperature have been added in line 94-96:

In order to prevent lipid oxidation, some phenolics was added to fish oil. The stripped fish oil was prepared in order to exclude the interference of the original phenolics in the fish oil.

At room temperature, the chromatographic silica was repeatedly rinsed with double-distilled water until it was free of impurities, and then activated at 120°C for 12 hours.

Point 20: L116: to which phase was sodium azide added?

Response 20: Added to the aqueous phase.

Point 21: Fig 1 is not needed at all. Just write the process clearly. That should be enough.

Response 21: It has been deleted.

Point 22: L121: what is a Fenton system? Give full details with ref.

Response 22: Information and references have been added in line 115-117:The emulsions were promotion oxidized under the Fenton system for 5 days, with samples collected every 24 h for analysis. The Fenton system was generated from a recovered solution of 10 µM FeCl3, 100 µM ascorbic acid, and 5 mM H2O2.

Yang, J.; Xiong, Y.L. Comparative time-course of lipid and myofibrillar protein oxidation in different biphasic systems under hydroxyl radical stress. Food Chem 2018, 243, 231-238, doi:10.1016/j.foodchem.2017.09.146.

Point 23: L 191: give ref #Give ref for ITC work.

Response 23: Added in line 179:Wang, Y.; Sun, Y.; Li, M.; Xiong, L.; Xu, X.; Ji, N.; Dai, L.; Sun, Q. The formation of a protein corona and the interaction with α-amylase by chitin nanowhiskers in simulated saliva fluid. Food Hydrocolloids 2020, 102, doi:10.1016/j.foodhyd.2019.105615.

Point 24: L243: This should not be oil. It should cream layer

:243-4: If you centrifuge 0.2 ml emulsion, how much cream and aq phase did you collect? Is that even possible? That seems too less.

Response 24: Here, 4 ml of the emulsion was centrifuged, and after centrifugation, 0.2 ml each of the oil and water phases were taken.Specific information in line 209-215:The sesamol emulsion (4 mL) was centrifuged (4°C, 10,000 g, 1 h) and the aqueous phase was carefully collected a syringe. Added 2 ml of PBS (10 mm, pH 7) to the re-mained emulsified layer, vortexed for 10 min, centrifuged (4°C, 3000 g, 5 min) to re-move the aqueous phase, and repeated 3 times, then added 1 ml of isooc-tane-isopropanol (3:1), vortexed for 10 min and centrifuged (4°C, 3000 g, 5 min) to re-move the lower aqueous phase and the intermediate emulsifier layer, and carefully collected the organic phase (oil phase) using a syringe.

Point 25:L245: 2 ml methanol was added to which phase and to much how much of that phase. Please clarify in the manuscript.

Response 25: Specific information has been added in line 215-219:The sample (emulsion, oil or aqueous phase) of 0.2 ml was vortexed in a 5 mL centrifuge tube with 2 mL of metha-nol (3 min) and then placed in an ultrasonic water bath (5 min) and the supernatant was collected and repeated 3 times. The supernatant was transferred to a dark glass vial to evaporate the solvent and fix the volume to 1 ml. Sample 1 ml was injected into the sample vial with an organic 0.22 µm filter.

Point 26:L273: why results and analysis? Why no results and discussion?

What os the pH of the emulsion? Please provide this data and use it to explain your results.

Response 26:

It has been changed to Results and Discussion.

The pH7 of the emulsion is higher than the isoelectric point of the protein and used to explain the negative charge of the droplet in line 266-267:The pH 7 of the emulsion is higher than the WPI isoelectric point, so the charges of the droplets are all negative.

Point 27:L278: here and all similar cases: Particle size should be replaced with droplet size

Response 27:All particle sizes have been replaced with droplet sizes.

Point 28:L294: clarify what is your control emulsion

Response 28: 0.00% sesamol emulsions were used as control samples and instructions have been added in line 249:Nevertheless, the sesamol-added emulsions had lower D4,3 values than the control sample (0.00% sesamol emulsions), and the average droplet size tended to decrease with the increase in sesamol content.

Point 29:L 296: it did not remain a lower value after 5 days

Response 29:Has been corrected in line 249: In addition, the oil droplet size of the control emulsion decreased on day 2 of storage and remained at a lower value from day 2 to day 4 .

Point 30: How did you know aggregation and also deoiling? It is not possible from your data here.

Response 30: It can be inferred from the smaller particle size.The larger the oil droplet size, the faster the rate of agglomeration. The newly prepared control emulsion has the largest droplet size and it will form a creamy layer or oil layer on top of the emulsion first because of flocculation or agglomeration of large size droplets. The proportion of small size droplets remaining in the emulsion is much larger, so the light scattering measurement reflects only the size of the small droplets remaining in the emulsion.

Point 31:L302: It did not inhibit on day 5. Please fix

Response 31: Has been corrected: inhibited the degree of droplet aggregation during the first four days of storage in line 258-260:The addition of 0.09% sesamol not only promoted the formation of small oil droplets during emulsion preparation, but also inhibited the degree of droplet aggregation dur-ing the first four days of storage.

Point 32:L307:What data showed that repulsive forces between the droplets increased? You do not have any proof. Please fix.

Response 32: It has been already deleted.

Point 33:L 320: you should show this by using just WPI and polyphenol without the emulsion

Response 33:WPI (1%, w/w) and sesamol (0%, 0.01%, 0.03%, 0.09%, w/w) were dissolved and hydrated with PBS (10 mm, pH 7), and the zate potential was measured after 200-fold dilution with PBS (10 mm, pH 7).The zate potential value is:-14.07、-14.27、-13.07、-13.67. It was demonstrated that the addition of 0.03% and 0.09% sesamol had a tendency to reduce the WPI zate potential.

Point 34:L 323-4: did they add the antioxidant to the emulsion? Or to oil or to the water? Please clarify

Response 34:Has been corrected in line 272-274: Our observations are in agreement with Yi et al. who found that the addition of the antioxidant black rice anthocyanin to the aqueous phase of walnut O/W nanoemulsion resulted in a decrease in the absolute value of ζ-potential.

Point 35:L338: How did you know deoiling. There is not clear proof. It might be just creaming.

Response 35: It has been already deleted.

Point 36:L340: how come oxidation leads to deoiling? It Is not clear at all.

The visual observation images are not entirely clear to understand any phase separation or the claimed deoiling.

Response 36:Corrected to in line 285-287:The products of the Schiff base reaction during the oxidation of the emulsion change its color to yellow, so the color change proves that the formation of the cream layer originates from the oxidation of lipids.

Point 37:L352-355: This is not clear from the figure. Better explanation and proof needed.

Response 37:Corrected to in line 295-297:Has been corrected:On day 5, all emulsions were observed to have droplet flocculation and coalescence, with the smallest droplets observed for the 0.09% sesamol emulsion as well as the least coalescence.

Point 38: L355: this result does not prove lipid oxidation

The first paragraph of Section 3.2 and similar sections: These first paragraphs are just repetitions of what you said in methods and are not needed. I suggest that you remove these from the manuscript. It would be better for reading.

Response 38: It has been already deleted.

Point 39:Fig 4: what is the x-axis here?

Fig 4 and similar fig caption: Need to give full details in the caption. What temp? what condition, the vials were stored etc.

Response 39:X-axis is storage time. The abscissa and storage information (30ml of emulsion in a 50ml glass bottle with screw cap stored at room temperature and protected from light) have been supplemented in Figure 4.

Point 40: Section 3.3: mechanism of protein oxidation should be discussed in more details.

Response 40:The principle of protein oxidation has been supplemented in line 344-347: The protein peptide backbone is attacked by reactive oxygen species to lose hydrogen atoms to form protein radicals, which react with oxygen to form peroxyl radicals, fol-lowed by a series of protein oxidation reactions. Sesamol can inhibit protein oxida-tion through its strong hydrogen supply capacity.

Point 41: L444-6: It is not clear how you calculated the KD and other values from the data. Give more details and ref.

Point 42: L449: You did not show how you got s<0

Response 41.42:The relevant calculation equations have been added in line 184-186:The unit point combination model was selected for data fitting and calculated accord-ing to the Van't Hoff equation:

                                 (2)

                                (3)

Point 43: L457-9: The explanation is not clear.

Response 44:It has been changed in line 385-387:The negatively charged O and N atoms on the protein polypeptide chain can form hydrogen bonds with the positively charged hydrogen atoms on the polyphenol phenolic hydroxyl group.

Point 44: Section 3.5: the first sentence is not needed.

Response 44: It has been already deleted.

Point 45: L502: The WPi films are not very thick. Please be cautious what you write.

Response 45: Has been corrected in line 423-425:This effect results from the ability of WPI to adsorb and form intercon-nected, viscoe-lastic films at the oil-water interface.

Point 46: L510: not clear what you mean by aggregation of Sesamol. Please explain better.

Response 46:Has been corrected in line 430-432:Hydrogen bonding, van der Waals forces, and hydrophobic interactions support and compete with each other to promote the proximity of sesamol to the WPI surface cavity binding site and its aggregation near the binding site.

Point 47: L514: this interaction with multiple proteins theory has no proof. Please discuss it with more details with solid evidence.

Point 48 :L 517: your data does not support this claim. Again, you cant say this type of speculation withoit actual data.

Point 49: L519-521: Did this ref actually show this with proof? How did you they do it> please give more details and explain better.

Response 47.48.49: Has been corrected in line 432-436:In addition, the further cross-linking of sesamol with the adsorbed layer WPI may form a synergistic mechanism. Dimitris et al. hypothesized that chlorogenic acid at the interface is able to form more hydrogen bonds with multiple adjacent protein mole-cules, inducing protein unfolding to form a more efficient interfacial coverage.

Point 50: Section 3.7: this first paragraph is not needed

Response 50:Already deleted.

Point 51: L 538: are you talking about one or multiple studies here?

Response 51:Has been corrected in line 445-446:The partitioning of sesamol in the aqueous phase in this study was lower than in previous studies.

Point 52:L554: not clear how you can say the tendency to migrate to the oil phase, when the Sesamol was added to the oil phase at the beginning. Did you entirely move to aq phase and then migrate back> Is that even possible? Also, see my comments on your wrong calculation of partitioning at the beginning. This must be redone.

Response 52:Has been corrected in line 452-454: Sesamol content in the oil phase increased significantly during the first 2 days of emulsion storage, indicating that sesamol in the aqueous phase and interfacial layer diffused into the oil phase, increasing the partition ratio of sesamol in the oil phase

Point 53:Your explanation in 557-567 is not clear. This must be rewritten based on better data.

Response 53:Has been corrected in line450-462 :After 5 days of storage, the total amount of emulsion sesamol was reduced to one-third of the original level, which was associated with chemical degradation during oxidation [53]. Sesamol content in the oil phase increased significantly during the first 2 days of emulsion storage, indicating that sesamol in the aqueous phase and interfa-cial layer diffused into the oil phase, increasing the partition ratio of sesamol in the oil phase [54]. After redistribution by diffusion, the partition ratio of the oil phase sesamol reached a maximum, while the partition ratio of the interfacial layer sesamol reached the lowest level.On the third day of storage, the sesamol content in both the oil and aqueous phases decreased by 50%, while the sesamol content in the interfacial layer increased, indicating that some of the sesamol in the oil and aqueous phases diffused into the interfacial layer in addition to oxidative degradation. In summary, the distri-bution of sesamol in the emulsion during storage is dynamic. We hypothesize that the micelle or vesicle structure in the emulsion facilitates the dynamic diffusion of sesamol between the phases

Point 54:L569-571: You can't say that. You only studied for 5 days. You don’t know what will happen after that time.

Response 54: Already deleted

Point 55:L578-580: This is not clear from the figure. Please fix.

Response 55: Already deleted

Point 56:L583-586: there is no proof of this statement.

Response 56: Already deleted

Round 2

Reviewer 1 Report

I only have one comment :
lines 106-107 : "(0, 0.01, 0.03 and 0.09% in the final emulsion, w/v) ". Not clear, please correct.

Author Response

Dear reviewer 1,

Thank you for your useful comments and suggestions on our manuscript. We have modified the manuscript accordingly, and detailed corrections are listed below point by point:

Point 1: (x) English language and style are fine/minor spell check required

Response: English language and style have been improved. Spell check has been performed as recommended.

Point 2: lines 106-107 : "(0, 0.01, 0.03 and 0.09% in the final emulsion, w/v) ". Not clear, please correct.

Response: Already modified according to the suggestions in lines 106-107 : The oil phase was prepared by adding sesamol (sesamol content in the emulsion was 0, 0.01, 0.03 or 0.09%, w/v) to the fish oil.

Reviewer 2 Report

The authors made a big effort in improving the mansucript and , in general, incorporated most of the suggestions made by the reviewers.

The authors should consider the following comment.

The authors themselves recognize that the distribution of sesamol obtained is biased because they break the emulsion prior to analyses. Eventhough this is also done by other authors, it does not mean that it is correct, an as so must be indicated in the paper.

Thus, the authors must indicate explicity in the manuscript that the distribution data may not reflect the real one because the break the emulsion prior to analyses. However, the lack of other methodologies makes them to make this tentative calculations. In equations 2-4, they must explicity indicate that the values provided by the equations are approximate. 

If there is any other method to calculate the actual concentrations of antioxidants (the authors mention the possibility of doing it by employing a chemical probe) then it must explicity indicated in the manuscript that the actual concentrations can be (potentially) determined by employing such methodology and should provide adequate references.

Author Response

Dear reviewer 2,

Thank you for your useful comments and suggestions on our manuscript. We have modified the manuscript accordingly, and detailed corrections are listed below point by point:

Point 1: The authors themselves recognize that the distribution of sesamol obtained is biased because they break the emulsion prior to analyses. Even though this is also done by other authors, it does not mean that it is correct, an as so must be indicated in the paper.

Thus, the authors must indicate explicity in the manuscript that the distribution data may not reflect the real one because the break the emulsion prior to analyses. However, the lack of other methodologies makes them to make this tentative calculations. In equations 2-4, they must explicity indicate that the values provided by the equations are approximate. 

If there is any other method to calculate the actual concentrations of antioxidants (the authors mention the possibility of doing it by employing a chemical probe) then it must explicity indicated in the manuscript that the actual concentrations can be (potentially) determined by employing such methodology and should provide adequate references.

Response

It has been pointed out in the text that the results of centrifugal measurements are biased and the results may be inaccurate in line 462-463:Accurately determining the interfacial distribution of antioxidants in emulsions is not a simple task. Separating different phases by centrifugation maybe disrupt the equilibrium of the interfacial region, and the interfacial distribution of antioxidants detected on this basis may not reflect the true situation.

The lack of alternative methods has been noted in the text in line 464-465:However, there is no method to determine the interfacial distribution of antioxidants by directly measuring the content of antioxidants in each phase.

It is already clear that the values provided by the equation are approximate in line 223-225:

          (4)

            (5)

            (6)

The method of chemical probes and 4 references have been described in line 466-469:The relatively scientific approach is based on the reaction between the 4-hexadecyl diazenium ion (16-ArN2+(BF4-)) molecular probe and the antioxidant, and the kinetic equations are used to calculate the partition constants and interfacial molar values of the antioxidant between different interfaces in the emulsion[59-62].
